# Multi-FAct: Assessing Factuality of Multilingual LLMs using FActScore

**Sheikh Shafayat, Eunsu Kim**\*, **Juhyun Oh**\*, **Alice Oh**
School of Computing
KAIST
Daejeon, Korea
{sheikh.shafayat,kes0317,411juhyun}@kaist.ac.kr,alice.oh@kaist.edu

## Abstract

Evaluating the factuality of long-form large language model (LLM)-generated text is an important challenge. Recently there has been a surge of interest in factuality evaluation for English, but little is known about the factuality evaluation of multilingual LLMs, specially when it comes to long-form generation. We introduce a simple pipeline for multilingual factuality evaluation, by applying FActScore (Min et al., 2023) for diverse languages. In addition to evaluating multilingual factual generation, we evaluate the factual accuracy of long-form text generation in topics that reflect regional diversity. We also examine the feasibility of running the FActScore pipeline using non-English Wikipedia and provide comprehensive guidelines on multilingual factual evaluation for regionally diverse topics.

## 1 Introduction

Large Language Models (LLMs) are susceptible to factuality hallucination, a phenomenon in which the generated text contradicts established world knowledge (Huang et al., 2023; Zhang et al., 2023). Despite extensive research focusing on hallucination and factuality of LLMs in free-form generation, the previous works have predominantly studied English (Huang et al., 2023; Min et al., 2023; Mishra et al., 2024; Wei et al., 2024; Wang et al., 2023). Consequently, there exists a wide gap in our comprehension of the factual accuracy of LLMs when producing content in non-English languages. As highlighted by Kang et al. (2024), current metrics for detecting hallucination are inadequate in multilingual settings. The extent to which LLMs exhibit factual hallucination remains unclear across different languages. Similar to other capabilities, there may be a discernible decline in performance when LLMs are tasked with non-English contexts (Ahuja et al., 2023; Bang et al., 2023).

In this paper, we address this gap by systematically evaluating the factual accuracy of multilingual LLMs across different languages and geographic regions. We explore the following research questions: **R1:** How can we effectively evaluate the factuality of multilingual LLMs in *free-form* text generation? **R2:** How do multilingual models' free-form answers compare in factual accuracy across languages and geographically contextualized questions?

To address these questions, we introduce Multi-FAct , a simple pipeline tailored for evaluating factuality in a multilingual context. Leveraging open-source models, we adapt the FActScore (Min et al., 2023) for multiple languages and validate it using human annotations from Min et al. (2023). Furthermore, we explore the use of non-English Wikipedia content to augment factuality evaluation, finding that while English Wikipedia remains a reliable source, concatenating articles from multiple non-English languages can sometimes provide estimates close to English Wikipedia.

We conduct an evaluation of the factual accuracy of recent proprietary multilingual LLMs in nine languages using biography generation task. We ensure geographical diversity by curating subjects from various regions. Our analysis reveals two significant findings. First,

---

\*Equal contribution.

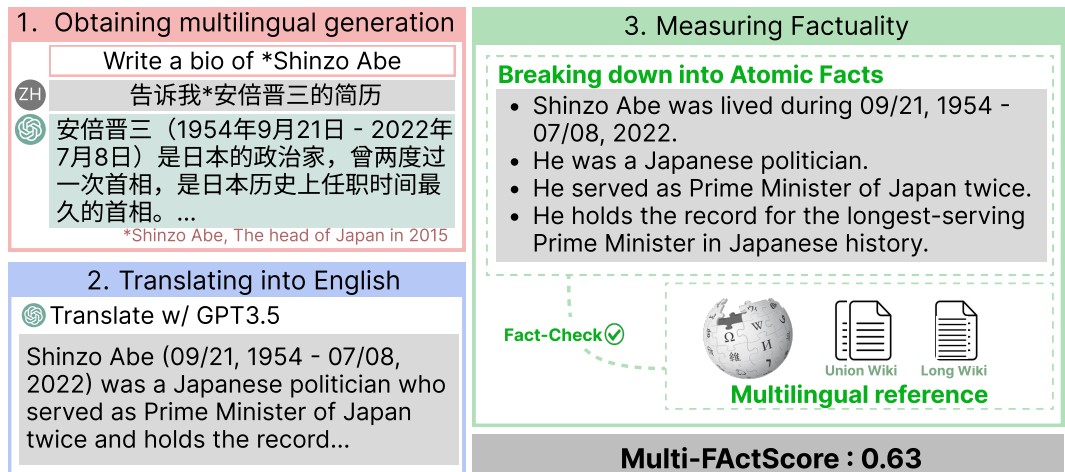

Figure 1: Our Multi-FAct Pipieline. The pipeline is structured into three main stages: 1) Obtaining multilingual generations, 2) Translating these facts into English using GPT-3.5, and 3) Measuring factuality, by breaking them down into smaller atomic facts (Min et al., 2023) and then verifying them by asking LLMs with context provided.

English consistently maintains an advantage in both factual accuracy and the quantity of generated facts compared to other languages. Second, content produced by multilingual language models exhibits better factual accuracy about North America and Europe across the languages.

Our contributions are as follows:

- We propose a novel pipeline Multi-FAct , that applies FActScore in a multilingual setting, and demonstrates the feasibility of conducting factuality evaluation of long-form generation in the multilingual biography task.

- We explore the potential and limitations of using non-English resources, particularly non-English Wikipedia articles, for factuality evaluation of long-form generation, showing that reasonably large non-English Wikipedia articles can also serve as a viable source.

- We introduce Multi-FAct as a tool to investigate cultural and geographic biases in LLMs, offering insights into how these biases manifest in LLM generated content.

## 2 Related Work

**LLM Factuality Evaluation**     The assessment of factual accuracy in natural language texts predates the advent of LLMs (Guo et al., 2022). Our Multi-FAct pipeline follows the previous fact evaluation research, which involves validating claims based on external resource such as Wikipedia article (Thorne et al., 2018; Krishna et al., 2022; Zhong et al., 2019) or Google search (Chern et al., 2023; Wei et al., 2024). Our work is inspired by and based on Min et al. (2023) which proposes FActScore to measure the factuality of a long-form generated output of LLMs by breaking it down into atomic facts. Another line of work focuses on evaluating factual accuracy solely based on model output or internal states, eliminating the need for external knowledge bases like Wikipedia (Manakul et al., 2023; Azaria & Mitchell, 2023; Dhuliawala et al., 2023). While such methods offer resource efficiency, they often sacrifice accuracy compared to approaches leveraging external resources for factuality evaluation.

**Multilingual Factuality Evaluation**     To evaluate factual accuracy of multilingual genera-tion, Gupta & Srikumar (2021) presents the X-Fact dataset. Similarly, Thorne et al. (2018)

presents frameworks for multilingual fact-extraction and verification evaluation. Additionally, Aharoni et al. (2022) introduces mFACE, a dataset aimed at evaluating multilingual factual consistency. While these papers advance the understanding of multilingual factuality, they focus on the assessment of factual accuracy in existing articles or benchmarks rather than examining the factual accuracy in the long-form generation of LLMs. Recently, Kang et al. (2024) explores the efficacy of existing metrics designed to detect hallucinations of LLM-generations in English in a multilingual setting, showing that the English-based metrics fall short in multilingual contexts. Our study extends the endeavor of developing a factuality evaluation metric that can be applied in multiple languages by introducing a Multi-Fact pipeline.

**Geo-culture biases of LLMs**   Research shows LLMs exhibit cultural and linguistic bias (Hovy & Yang, 2021; Cao et al., 2023; Hershcovich et al., 2022; Huang & Yang, 2023; Jin et al., 2023; Naous et al., 2023). A recent paper conducts a true-false evaluation task that shows GPT models are more accurate for facts about the Global North compared to the Global South (Mirza et al., 2024), consistent with our findings. We conduct a much more comprehensive study with long-form generation of geographically diverse information in multiple languages.

# 3   Multi-FAct Pipeline

We introduce Multi-FAct, a novel pipeline for automatically measuring factuality of a multilingual LLM in a multilingual setting. Multi-FAct evaluates the biographies of multi-regional topics in multiple languages. We choose the biography generation task because biographies are suitable for being decomposed into verifiable and independent "atomic" facts (Min et al. (2023)).

The Multi-FAct pipeline is structured into three main steps: 1) Obtaining multilingual generations (§ 3.1), 2) Translating these facts into English using GPT-3.5 (§ 3.2), and 3) Measuring factuality of the translated facts (§ 3.3).

## 3.1   Generating Facts

**Model**   Most of the results in this paper use GPT4 (`gpt-4-1106-preview`) and GPT3.5 (`gpt-3.5-turbo-0613`) models because at the time of writing this paper, those were the only two strong multilingual models[1] for LLM multilingual biography generation. All experiments are conducted between January and March 2024[2], and the model temperatures are set to their defaults for respective models.

**Language**   We choose a total of nine languages: English, German, French, Spanish, Arabic, Swahili, Chinese, Korean, and Bengali. We carefully choose languages to represent the level of existing resources–high, medium, and low, and to represent diverse regions–Africa, Europe, Asia, and America [3].

**Prompt**   We use the following prompt: `Write a biography of {name}`. We translate the prompt with GPT-4 into the eight non-English langauges, then we ask the native speakers to verify the translation. We also transliterate each person's name into corresponding languages either using Wikipedia crosslinking or GPT-4, of which we take a subset and the native speakers verify that almost all the transliterations are correct.

---

[1]We also provide FActScore estimation for some newer models in Appendix C.

[2]Except for GPT4o and GPT4o-mini in the Appendix.

[3]**Africa:** Arabic, Swahili, and French; **Europe:** Spanish, French, German; **Asia:** Korean, Bengali, Chinese; **America:** Spanish and French. English is widely spoken all over the world, and kept as the baseline.

## 3.2 Translating Generated Facts

To compare the biographies written in the eight non-English languages, we translate the generated content from the original languages to English using `gpt-3.5-turbo-0125`. We do not use commercially available machine translation systems, such as Google Translate, based on the preliminary result that they do not maintain the consistent gender of the person throughout the text[4].

We translate the generation and verify facts in English, rather than doing fact verification in corresponding languages with respective Wikipedia articles for multiple reasons. First, Wikipedia differs in size and scope for each language (Wikipedia contributors, 2024), which makes it difficult to compare cross-lingual factuality if the knowledge base is different. Additionally, key components of the original FActScore pipeline, such as RAG and NPM, are optimized for English and not available in other languages. Our quantitative analysis of LLM-based translation's impact on FActScore evaluation reveals that GPT-3.5's translation minimally influences the FActScore of texts (See § 3.4).

## 3.3 Measuring Factuality

To evaluate the accuracy of the generated model's response $M$, we use the FActScore metric. This involves breaking down the translated generation into atomic facts—short sentences, each conveying a single piece of information, as defined in Min et al. (2023). We denote the set of atomic facts as $A$, and we measure the accuracy of these atomic facts with the corresponding English Wikipedia article serving as the reference knowledge source $C$.

$$\text{\# of Correct Facts}(M) = \mathbb{1}(\text{a is supported by } C)$$

$$\text{FActScore } (M) = \frac{1}{|A|} \sum_{a \in A} \text{\# of Correct Facts}(M) \tag{1}$$

We replace all proprietary models of the original FActScore pipeline with open-source models, ensuring cost-effectiveness while maintaining the quality of our system. We decompose LLM responses $M$ into atomic facts using Mistral-7B Jiang et al. (2023). The verification step also uses Mistral-7B along with RAG and NPM for the best performance. We set the NPM threshold at 0.3.

## 3.4 Reliability of Multi-FAct

### 3.4.1 Replication of original FActScore

We conduct a comparative analysis using our implementation, which includes a Mistral 7B model, instead of InstructGPT for atomic break-down, against the outcomes presented in the Min et al. (2023), using a subset of human-annotated factual data generated by ChatGPT (See Table 1). A comparison with other baselines and different parts of the Multi-FAct pipeline for different languages is provided in appendix A.

| Metric | Human FS | FS (ours) | Error |
|--------|----------|-----------|-------|
| Value  | $0.626 \pm 0.238$ | $0.640 \pm 0.127$ | $-0.014 \pm 0.128$ |

Table 1: Summary of Replication. Error refers to the difference between human FActScore and FActScore determined by our method. Standard deviation is reported alongside each metric. Note that the Human FActScore value is slightly different from what is provided in Min et al. (2023) as this value is averaged over our subset (44 samples).

---

[4]However, we do provide an analysis of what happens when Google Translation is used instead of GPT3.5 in Appendix C.

### 3.4.2 Effect of GPT-3.5 Translation in Multi-FAct

As our pipeline crucially relies on the GPT-3.5 translation (Step 3 in figure 1), it is imperative to check whether the translation step affects the automated estimation of FActScore in the multilingual setting. We evaluate the effect of GPT-3.5 translation based on the premise that the factual content of a text should remain consistent across translations into different languages. In other words, if the factuality of an English text is established, its translated version in another language should exhibit the same level of factuality.

| Language | Correlation | Mean error | Standard Deviation |
|----------|-------------|------------|--------------------|
| English  | 0.84        | -0.014     | 0.13               |
| German   | 0.83        | -0.007     | 0.13               |
| French   | 0.80        | -0.004     | 0.14               |
| Spanish  | 0.81        | -0.014     | 0.14               |
| Swahili  | 0.81        | -0.015     | 0.13               |
| Arabic   | 0.82        | 0.021      | 0.14               |
| Chinese  | 0.81        | 0.040      | 0.14               |
| Korean   | 0.80        | 0.009      | 0.14               |
| Bengali  | 0.84        | 0.027      | 0.13               |

Table 2: FActScore evaluation across languages. English refers to FActScore evaluation on original human annotation data provided in English; other languages refer to Multi-FAct evaluation on GPT-4 translation of the human annotation. Correlation refers to the correlation of individual FActScore with human-annotated FActScore provided in Min et al. (2023).

To simulate and test the reliability of Multi-FAct with respect to translation, we first take the human-annotated examples provided in the original work (Min et al., 2023) and translate them into the eight non-English languages using GPT-4[5]. Then, we apply our multilingual FActScore pipeline (Figure 1) on the translated texts[6]. Table 2 indicates that the execution of Multi-FAct on non-English data does not significantly degrade the estimation of FActScore, and the correlation between FActScore in different languages and human evaluation remains comparable. This outcome implies that adding a translation step to our pipeline does not significantly influence the estimation of FActScore .

## 4 Reference Sources in Multilingual Factuality Evaluation

We evaluate the factual accuracy of LLM-generated biographies against English Wikipedia for its reliability and comprehensive coverage of information about individuals, following prior works (Petroni et al., 2020; Chen et al., 2017; Min et al., 2023). However, we need to analyze the reliability of non-English Wikipedia articles as references. While the Multi-FAct pipeline is effective for topics with established English Wikipedia articles, its applicability may be limited because many regional topics (e.g., local politicians) lack English Wikipedia articles. If we cannot extend our Multi-FAct pipeline to these topics, it would be a major obstacle to ensuring LLM factuality with geographical diversity. However, as figure 2 illustrates, languages included in our study exhibit a wide range of variability in their representation within Wikipedia.

In this section, we explore the following question: How should we measure multilingual factuality, when we cannot find a sufficient golden source of reference *in* English? First, we examine whether simple translation of Wikipedia articles from languages other than English would sustain the quality (§ 4.1). Then, we reveal that the "length of articles" is one factor contributing to performance degradation when non-English Wikipedia is used as fact verification ground truth corpus, and we propose a new method of concatenating articles from different languages to enhance the comprehensiveness of the reference (§ 4.2).

---

[5]We use GPT-4 as a previous study (Jiao et al., 2023) demonstrated GPT-4 to be a good translator.
[6]We did not use GPT-4 for back-translation in Multi-FAct pipeline to reduce API costs.

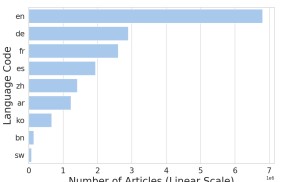
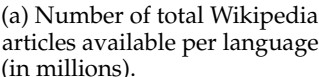

(a) Number of total Wikipedia articles available per language (in millions).

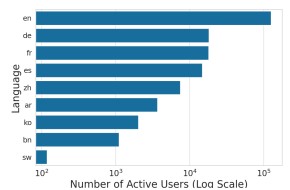

(b) Number of active Wikipedia users per language (note the log scale).

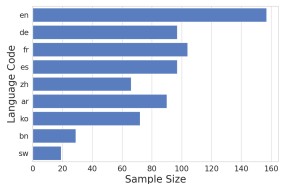

(c) Number of original FActScore annotated topics that have Wikipedia articles in corresponding languages.

Figure 2: The Wikipedia size distribution for languages in this study. (c) shows the number of human-annotated examples that also have representation in the corresponding Wikipedia. Note that Wikipedia size differs widely across languages, which usually means non-English Wikipedia articles are not as comprehensive as English Wikipedia articles for fact verification.

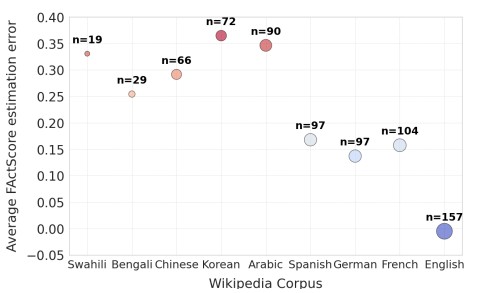
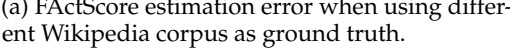

(a) FActScore estimation error when using different Wikipedia corpus as ground truth.

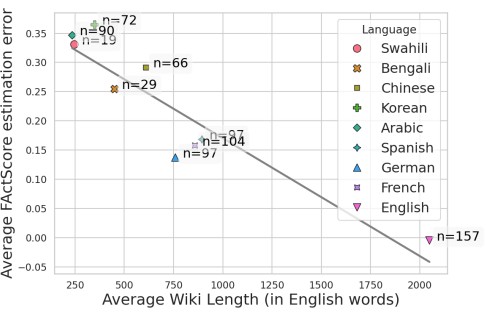

(b) FActScore estimation error scales linearly with the length of the Wikipedia article.

Figure 3: Effect of using non-English Wikipedia as knowledge corpus for FActScore estimation. The x-axis in subfigure 3b represents the word count (in English) after translation. As the length of the non-English Wikipedia article increases, its utility for evaluating factuality increases.

## 4.1 Using Translated Wikipedia Articles as Reference

We first test whether we can simply translate the Wikipedia articles written in eight non-English languages into English then replace the English Wikipedia articles within Multi-FAct pipeline with the translations. Using GPT-3.5, we translate the respective Wikipedia articles into English and conduct factual evaluation in English. We chose 157 individuals as topics for whom human-annotated data exists in Min et al. (2023), which served as the ground truth. We then compare this ground truth with the experimental results.

Figure 3a illustrates the errors for each language's Wikipedia against ground truth, and we include English Wikipedia corpus as the baseline. The results show a near-zero error rate for English, and lower error rates in French, Spanish, and German. We observe relatively higher error rates for Bengali, Korean, Chinese, and Arabic. Note that the size and coverage of Wikipedia varies across languages, thus the number of samples used in the experiments also varies by language, as indicated by $n$ in the figures.

## 4.2 Effect of Reference Length

We hypothesize that non-English Wikipedias result in higher errors compared to English Wikipedia due to the lower coverage of topics, as evidenced by the shorter length of non-

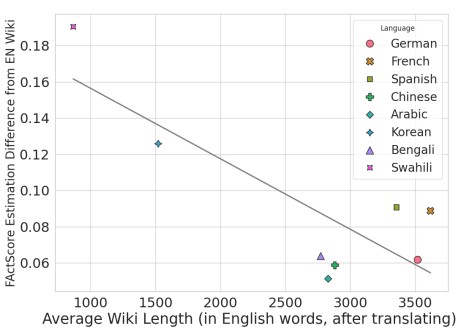

(a) Multi-FAct estimation for persons with 'large' Wikipedia articles in corresponding language.

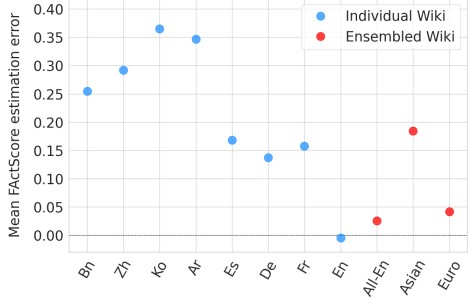

(b) Ensembling multiple non-English Wikipedias work better than single language Wikipedia.

Figure 4: Effect of corpus size on FActScore Estimation. Figure 4a shows that comprehensive non-English sources might be used for factuality evaluation after translating into English.

English articles. Figure 3b shows that there is a linear relationship between FActScore estimation error and the word count of Wikipedia articles, indicating that longer Wikipedia articles tend to serve better as references for factuality evaluation.

To further test the linear relationship observed in Figure 3b, we select 100 individuals from each of our studied languages who have both English Wikipedia articles and substantially long Wikipedia articles in their regional language.[7] We generate biographies for these 100 individuals in English and evaluate these biographies against both the English Wikipedia and the Wikipedia in the corresponding regional language. Since FActScore estimation for English is nearly perfect, we use it as a benchmark to estimate the error when using non-English articles as references. Figure 4a illustrates that the linear relationship between the length of Wikipedia articles and FActScore holds for longer articles as well.

**Ensembling Wikipedia** Building upon the insights about the reference length discussed in § 4.1, this section investigates whether combining multiple non-English Wikipedias, referred to as "ensembling Wikipedias," can enhance the accuracy of Multi-FAct measurements. In our experiments, we explore three cases: the union of wikis from Asian languages (Asian), the union of wikis from European languages (Euro), and the union of all wikis except English (All-Eng). As depicted in the Fig 3a, among the two wikis with errors close to zero—namely, "Euro" and "All-Eng" cases—it is speculated that this could be attributed to their comprehensiveness or the high content overlap with the English Wikipedia. In the case of "Asian," an error rate of 0.2 is observed, representing a decrease compared to using only one wiki of Asian language, and even approaching the performance observed for European languages (Spanish, French, and German).

Our findings suggest that even for the same topic, the coverage of Wikipedia articles may vary across languages, thereby demonstrating that an "ensemble Wikipedia" comprising multiple articles can mitigate the performance gap between non-English Wikipedias and the English Wikipedia. However, the challenge of Wikipedia containing only a single, brief article remains unaddressed.

## 5 Factual Accuracy of Multilingual models

We use Multi-FAct to investigate the factual accuracy of multilingual LLMs in biography generation. We select "National Leaders of Each Country" as the topic for factuality measurement, as it is fairly common across various cultures and languages to have well-known and well-described national leaders such as presidents and prime ministers. Specifically, we choose to focus on the president or head of state in the year 2015. This year is randomly

---

[7]The detailed sampling method is provided in appendix C.2

selected from those prior to the emergence of LLMs and also to make sure that at least some amount of data about the leader is online, which might not be true for very recent leaders. The lists of countries and their corresponding leaders used in our experiments are detailed in Table 8 of Appendix.

## 5.1 Does factuality differ across languages?

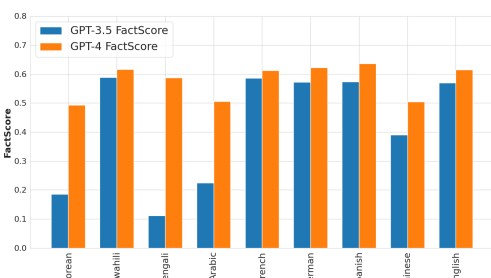

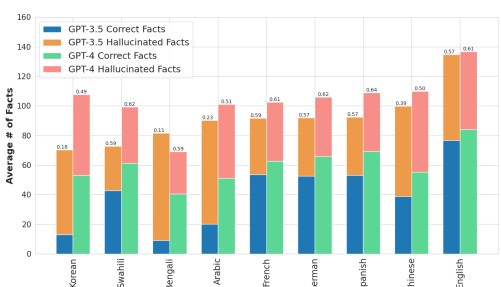

Figure 5: FActScore of GPT-3.5 and GPT-4 for each language. Note that GPT-4 makes a strong jump from GPT-3.5, especially in low-resource languages.

Figure 6: Number of correct and hallucinated facts by GPT-3.5 and GPT-4 for each language. The number on top of each bar is the FActScore .

Figure 5 illustrates the factuality across various languages using FActScore as a metric, uncovering significant variations among languages. English, Spanish, French, German, and Swahili exhibit notably higher FActScore for both GPT-3.5 and GPT-4.

Figure 6 compares factuality across languages through the average number of correct and hallucinated facts, highlighting a significant gap in the quantity of facts generated between English and other languages. Notably, even with similar FActScore as shown in Figure 5, English surpasses other languages in generating a higher number of correct facts, thus delivering more useful responses.

These observations emphasize the influence of output length differences on the number of correct and hallucinated facts across languages, despite similar FActScore values. English and other high-resource languages (e.g., Chinese, Spanish) typically produce more facts due to longer outputs. In contrast, low-resource languages like Bengali yield fewer facts because of shorter responses, even when their FActScore are comparable to English (0.58 for Bengali, 0.61 for English). This discrepancy further widens the factuality gap between low-resource and high-resource languages. Hence, the denormalization of output length in multilingual contexts suggests that evaluating factuality requires considering both FActScore and the number of correct facts for a thorough assessment.

## 5.2 Does the Factuality *Geographically* Differ Across Languages?

In this section, we explore whether the factuality of models varies with the geographic distribution of the language. Since GPT-4 performs better in low resource language (Figure 5), from here on, we conduct our analysis using only the outputs from GPT-4.

Table 3 presents the overall FActScore by language and continent. Interestingly, languages with higher mean values tend to have lower standard deviations, indicating that these languages maintain a relatively uniform level of factual accuracy regardless of the geographical area. Western languages such as English, Spanish, and German notably score high across continents, while Chinese, although a high-resource language, exhibits low factual precision. Additionally, among the nine languages evaluated, six demonstrate their highest performance in content related to the American continent, highlighting a prevalent American-centric bias in GPT's knowledge across most languages.

Next, we analyze the geographic distribution of the most factually accurate information across different languages. For each language, we identify the continents associated with

| Language | Africa | America | Asia | Europe | Mean | STD |
|---|---|---|---|---|---|---|
| Spanish | 0.624 | **0.641** | 0.627 | 0.640 | 0.633 | 0.087 |
| German | 0.622 | **0.642** | 0.616 | 0.628 | 0.627 | 0.094 |
| Swahili | **0.643** | 0.637 | 0.585 | 0.610 | 0.619 | 0.096 |
| English | 0.607 | **0.632** | 0.613 | 0.607 | 0.615 | 0.086 |
| French | 0.595 | **0.655** | 0.609 | 0.594 | 0.613 | 0.110 |
| Bengali | 0.579 | 0.574 | 0.585 | **0.589** | 0.582 | 0.149 |
| Arabic | 0.493 | **0.559** | 0.485 | 0.522 | 0.515 | 0.167 |
| Korean | 0.481 | **0.501** | 0.490 | 0.476 | 0.487 | 0.208 |
| Chinese | 0.453 | 0.502 | 0.479 | **0.514** | 0.487 | 0.230 |

Table 3: Overall FActScore breakdown of GPT4 by language with mean and standard deviation (STD). The continents with the highest and second-highest values for each language are emphasized in bold and underlined, respectively. Note that for all language except for Swahili, GPT4 is leaning towards American and European leaders in terms of factual accuracy.

top-20 FActScore . Our findings reveal a strong bias towards American and European topics across all languages studied. This pattern persists even in languages like geographicallly distant from these continents, such as Korean and Chinese. For Arabic, while the highest FActScore is linked to Africa, likely reflecting its geographical roots, American topics still rank second place. This trend highlights a Western-centric bias in the model's factual content distribution across languages. The result figure and detailed analysis of geographical biases across more fine-grained subregions are presented in Appendix D.

# 6   Conclusion

In this paper, we introduce Multi-FAct , an extension of FActScore in multilingual setting to address the critical gap in factuality evaluation for non-English free-form generation from LLMs. We empirically demonstrate the viability of conducting factuality evaluation in the multilingual biography generation task. Our approach of translating generated content and comparing it against English Wikipedia proved effective, offering a scalable method for assessing factual precision across multiple languages. Additionally, we find that sufficiently comprehensive non-English sources can serve as alternatives to English sources using the same pipeline.

Our analysis of LLMs' outputs across different languages and geographically contextualized questions reveals a consistent advantage for English in terms of both factual accuracy and the quantity of generated facts, as well as better performance for content related to North America and Europe across languages. Multi-FAct serves as a valuable tool for investigating these biases, emphasizing the need for efforts to improve the cultural and geographic fairness of factual generations from LLMs.

# 7   Limitations

Our work shares some of the same limitations that original FActScore faces. Throughout the whole paper, we assume English Wikipedia as the most authoritative knowledge base for fact verification. However, it is known that Wikipedia itself has a Western-centric bias Naous et al. (2023). Moreover, when combining multiple non-English Wikipedia, we do not take into account when Wikipedia articles in different languages contradict each other, which can happen for controversial topics. It is also worth noting that FActScore and Multi-FAct both do not differentiate between the informational value of facts. For example, statements like "Person X attended MIT" and "Person X attended a renowned university" are treated equally, despite the former being more specific and informative. Future research should aim to distinguish between specific, valuable facts and generic, less informative ones.

## Acknowledgments

This research project has benefitted from the Microsoft Accelerate Foundation Models Research (AFMR) grant program through which leading foundation models hosted by Microsoft Azure along with access to Azure credits were provided to conduct the research. We thank Pang Wei Koh and Sewon Min for providing us with valuable feedback on an earlier draft of this work.

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

## A  Comparison with Baseline

We provide a comparison of different components of Multi-FAct below.

**Trivial baselines:** Random prediction: 8.86% Always "Supported": - 41.15% Always "Not Supported": 58.86%

| Method | Arabic | Swahili | Bengali | Chinese | Korean | French | German | Spanish | Mean |
|---|---|---|---|---|---|---|---|---|---|
| No Retrieval ChatGPT | -0.268 | -0.282 | -0.260 | -0.265 | -0.277 | -0.278 | -0.288 | -0.290 | -0.276 |
| Retrieval + ChatGPT | -0.163 | -0.190 | -0.165 | -0.168 | -0.177 | -0.177 | -0.175 | -0.187 | -0.175 |
| Retrieval + Mistral | -0.150 | -0.180 | -0.169 | -0.162 | -0.172 | -0.171 | -0.175 | -0.174 | -0.169 |
| NPM | -0.097 | -0.133 | -0.105 | -0.096 | -0.098 | -0.140 | -0.149 | -0.153 | -0.121 |
| Retrieval + Mistral + NPM | 0.028 | -0.016 | 0.022 | 0.029 | 0.006 | -0.013 | -0.013 | -0.012 | **0.004** |

Table 4: Multi-FAct pipeline component comparison across different languages.

In Table 4:

- ChatGPT refers to GPT3.5-turbo.
- No Retrieval ChatGPT means simply asking ChatGPT whether a fact is correct or not (closed book setting).
- Retrieval + ChatGPT implies retrieving relevant passages from Wikipedia and then answering whether a fact is correct or not using GPT3.5.
- Retrieval + Mistral refers to retrieving passages and then using Mistral 7B to answer whether a fact is correct or not. This is similar to the previous setting except for the fact verification LLM is being switched to an open-source model.
- NPM refers to Non-Parametric Model (Min et al. (2022)), which requires having access to the Wikipedia article (ie, retrieval enabled).
- The final method is the ensemble version of NPM and Retrieval + Mistral and it performs the best (also reported in original FActScore work (Min et al. (2023))). This is the one we used throughout our paper for other analyses.

While GPT3.5 and Mistral 7B are very similar at atomic fact breakdown, we use Mistral 7B in all experiments to save cost.

## B  Effect of non-LLM Based Translation

In section 3.4.2, we showed that GPT3.5-based translation does not hurt the FActScore estimate. We have also explained the reason we use GPT3.5 over commercial translation systems like Google Translate is because Google Translate is not able to keep the gender of the subject consistent across the generation. Table 5 provides a comparison of translation of FActScore estimation when the translation is done by Google Translate vs GPT3.5.

The experiment set up for Table 5 was the same as Table 2. The small difference in Error with GPT3.5 column arises from the run-to-run stochasticity of the FActScore estimation.

**Can we skip the translation step altogether?**

We also experimented with generating atomic facts in the target language (thus, skipping the translation step) with French and Korean using GPT3.5 Turbo. Qualitative analysis with native Korean speakers revealed that the generated atomic facts are quite reasonably good. However, because there is no NPM model available for Korean and French, the FActScore estimates using atomic facts generated with those methods do not match the best-performing method (Retrieval+Mistral+NPM) provided in the above table. Moreover, tokenizer fertility for non-English languages is often much worse than for English, which makes it much more computationally expensive to do atomic fact breakdown in Korean. At the time of writing this paper, no small open-source model, like Mistral 7B, could perform atomic fact breakdown in a diverse multilingual setting.

| Source Language | GPT3.5 Error | GTranslate Error |
|---|---|---|
| Arabic | 0.028 | 0.051 |
| Swahili | -0.0157 | 0.0168 |
| Bengali | 0.0219 | 0.057 |
| Chinese | 0.0289 | 0.0883 |
| Korean | 0.0063 | 0.0453 |
| French | -0.0126 | -0.0106 |
| German | -0.0127 | -0.0019 |
| Spanish | -0.0115 | 0.0581 |
| **Mean** | **0.0041** | **0.0380** |

Table 5: FActScore estimation error rate comparison when translation to English was done using GPT-3.5 and GTranslate across different languages

# C    Additional Results

## C.1    Results from other models

Table 6 shows the benchmark results from Claude-3 and Gemini-1.0-pro model. We chose to use GPT4 for most of our analysis because GPT4 has the best performance in low resource languages.

| Language | (Claude) haiku | opus | sonnet | gemini-1.0-pro | gpt3.5 | gpt4-turbo | gpt4o-mini | gpt4o |
|---|---|---|---|---|---|---|---|---|
| English | 0.68 | 0.70 | 0.62 | 0.44 | 0.57 | 0.62 | 0.54 | 0.58 |
| French | 0.65 | 0.64 | 0.60 | 0.42 | 0.59 | 0.61 | 0.56 | 0.58 |
| Spanish | 0.66 | 0.70 | 0.60 | 0.47 | 0.57 | 0.63 | 0.59 | 0.59 |
| German | 0.63 | 0.66 | 0.58 | 0.43 | 0.57 | 0.63 | 0.55 | 0.60 |
| Chinese | 0.27 | 0.43 | 0.19 | 0.40 | 0.40 | 0.49 | 0.49 | 0.53 |
| Korean | 0.39 | 0.45 | 0.27 | 0.40 | 0.18 | 0.48 | 0.49 | 0.52 |
| Arabic | 0.33 | 0.45 | 0.26 | 0.32 | 0.24 | 0.53 | 0.46 | 0.51 |
| Bengali | 0.26 | 0.36 | 0.18 | 0.40 | 0.12 | 0.58 | 0.48 | 0.58 |
| Swahili | 0.57 | 0.62 | 0.54 | 0.46 | 0.59 | 0.62 | 0.65 | 0.65 |
| Mean | 0.49 | 0.56 | 0.43 | 0.42 | 0.43 | **0.58** | 0.53 | 0.57 |

Table 6: FActScore estimation of the other models on the same national leaders' biography generation task. Note that only GPT4-turbo has the overall highest performance and on average GPT4 series models have good performance across all languages.

Note that model response size varies significantly across languages and models, which is not captured in this table. Similar to what is shown in figure 6, most of the models in this table tend to generate much longer responses for English and high-resource languages. The average response length also differs from model family to model family.

## C.2    Effect of Reference Length

**Sampling Method**    We randomly collected 10,000 people from 9 different Wikipedia using Wikipedia API and sorted them by their byte length. After that, we kept the top 100 entities that also had an English Wikipedia. Note that since we did this for each language separately, the list of 100 people differs for each language. However, manual inspection revealed that these entities are usually famous politicians, religious figures, famous athletes, and singers.

Table 7 shows the word count distribution for the selected Wikipedia articles.

| Language | Mean Text Length (Language Wiki) | Mean Text Length (English Wiki) |
|---|---|---|
| Spanish | 3353.94 | 6895.68 |
| French | 3617.13 | 6569.71 |
| German | 3519.03 | 5723.78 |
| Korean | 1520.79 | 5144.43 |
| Chinese | 2879.44 | 4937.70 |
| Arabic | 2829.48 | 5648.35 |
| Swahili | 865.85 | 5003.25 |
| Bengali | 2772.49 | 5829.74 |

Table 7: Comparison of mean text lengths (in words) in Various Language Wikipedia after translating to English vs length of the same articles English Wikipedia. For the same subjects, English Wikipedia has much larger articles than the other languages.

# D  Geographical Biases in Factual Precision Across Subregions

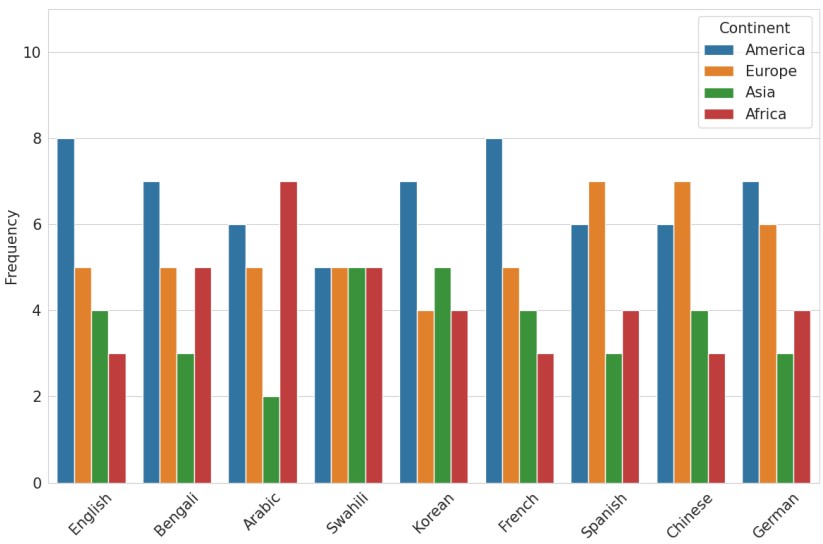

Figure 7: Continental distribution of top 20 countries that had the highest FActScore in each language for GPT4. The way to interpret this plot is, say for German, we sort the twenty most accurate presidents' biographies and then look up which continents they belong to. Note that for almost all languages, the top twenty biographies belong to subjects from America and Europe.

We examine how languages with notable variations in factuality across continents exhibit distinct geographical biases. Further analysis involves breaking down the continents into sub-regions to compare the number of correct facts and FActScore among three languages: Chinese and Korean, which exhibit he highest standard deviation (SD), and English, which demonstrates the lowest SD among the four continents, as detailed in Table 3 of § 5.2.

Chinese stands out in Eastern Asia, achieving the highest number of correct facts among all regions analyzed, indicating a pronounced bias towards its main linguistic area. Korean, also categorized in Eastern Asia, shows comparable levels of factuality in Eastern Asia to that of regions associated with America and Europe, underscoring geographical biases in the model's factual precision. English, while displaying a bias towards North America, its main region, exhibits relatively even factuality across other regions.

Figure 8 shows the fine-grained geographic distribution of languages with the highest standard deviation (Chinese, Korean, Arabic) and the least standard deviation (English).

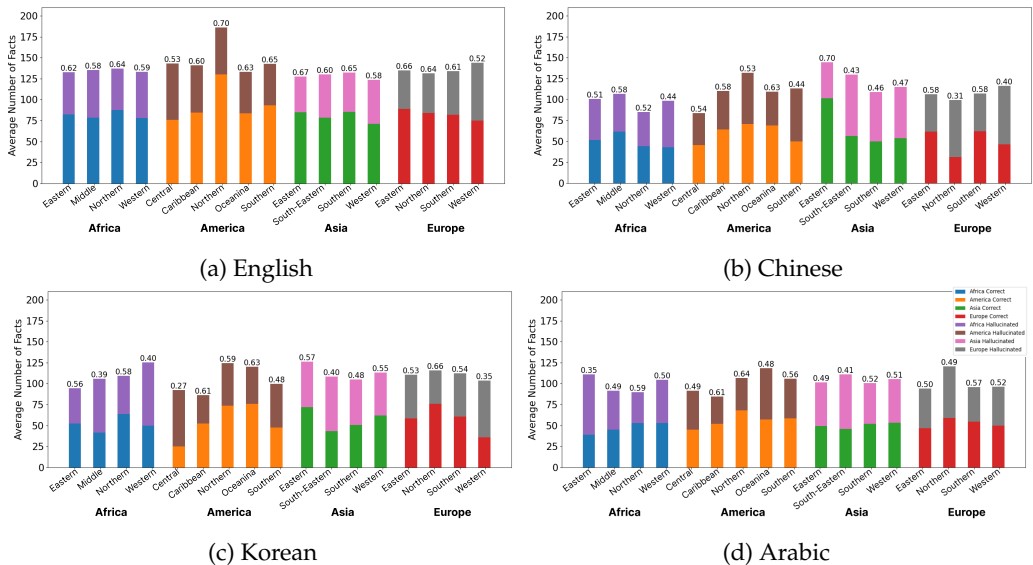

(a) English

(b) Chinese

(c) Korean

(d) Arabic

Figure 8: Fine-grained geographic distribution of languages with the highest standard deviation (Chinese, Korean, Arabic) and the least standard deviation (English) is given. The top bar represents the average number of hallucinated facts, and the bottom one denotes correct facts. The number on top of each bar represents FActScore .

# E   List of Countries and Presidents

Table 8 lists all the names of the countries, presidents and their Geographic locations. Please note that, we changed Australia and New Zealand to Oceania in the UN geoscheme for plotting purposes.

| Country | Continent | Region | President |
|---|---|---|---|
| Ethiopia | Africa | Eastern Africa | Hailemariam Desalegn |
| Tanzania | Africa | Eastern Africa | Jakaya Kikwete |
| Kenya | Africa | Eastern Africa | Uhuru Kenyatta |
| Uganda | Africa | Eastern Africa | Yoweri Museveni |
| Mozambique | Africa | Eastern Africa | Filipe Nyusi |
| Madagascar | Africa | Eastern Africa | Hery Rajaonarimampianina |
| DR Congo | Africa | Middle Africa | Joseph Kabila |
| Angola | Africa | Middle Africa | José Eduardo dos Santos |
| Cameroon | Africa | Middle Africa | Paul Biya |
| Egypt | Africa | Northern Africa | Abdel Fattah el-Sisi |
| Algeria | Africa | Northern Africa | Abdelaziz Bouteflika |
| Sudan | Africa | Northern Africa | Omar al-Bashir |
| Morocco | Africa | Northern Africa | Abdelilah Benkirane |
| South Africa | Africa | Southern Africa | Jacob Zuma |
| Nigeria | Africa | Western Africa | Muhammadu Buhari |
| Ghana | Africa | Western Africa | John Mahama |
| Côte d'Ivoire | Africa | Western Africa | Alassane Ouattara |
| Niger | Africa | Western Africa | Mahamadou Issoufou |
| Burkina Faso | Africa | Western Africa | Michel Kafando |
| Mali | Africa | Western Africa | Ibrahim Boubacar Keïta |
| Australia | America | Australia and New Zealand | Tony Abbott |
| New Zealand | America | Australia and New Zealand | John Key |
| Haiti | America | Caribbean | Michel Martelly |
| Cuba | America | Caribbean | Raúl Castro |
| Dominican Republic | America | Caribbean | Danilo Medina |
| Mexico | America | Central America | Enrique Peña Nieto |
| Guatemala | America | Central America | Otto Pérez Molina |
| Honduras | America | Central America | Juan Orlando Hernández |
| Nicaragua | America | Central America | Daniel Ortega |
| United States | America | Northern America | Barack Obama |
| Canada | America | Northern America | Justin Trudeau |
| Brazil | America | South America | Dilma Rousseff |
| Colombia | America | South America | Juan Manuel Santos |
| Argentina | America | South America | Cristina Fernández de Kirchner |
| Peru | America | South America | Ollanta Humala |
| Venezuela | America | South America | Nicolás Maduro |
| Chile | America | South America | Michelle Bachelet |
| Ecuador | America | South America | Rafael Correa |
| Bolivia | America | South America | Evo Morales |
| Paraguay | America | South America | Horacio Cartes |
| Uzbekistan | Asia | Central Asia | Islam Karimov |
| China | Asia | Eastern Asia | Xi Jinping |
| Japan | Asia | Eastern Asia | Shinzo Abe |
| South Korea | Asia | Eastern Asia | Park Geun-hye |
| Indonesia | Asia | South-Eastern Asia | Joko Widodo |
| Philippines | Asia | South-Eastern Asia | Benigno Aquino III |
| Vietnam | Asia | South-Eastern Asia | Trng Tn Sang |
| Thailand | Asia | South-Eastern Asia | Prayut Chan-o-cha |
| Myanmar | Asia | South-Eastern Asia | Thein Sein |
| Malaysia | Asia | South-Eastern Asia | Najib Razak |
| India | Asia | Southern Asia | Narendra Modi |
| Pakistan | Asia | Southern Asia | Mamnoon Hussain |
| Bangladesh | Asia | Southern Asia | Sheikh Hasina |
| Iran | Asia | Southern Asia | Hassan Rouhani |
| Afghanistan | Asia | Southern Asia | Ashraf Ghani |
| Nepal | Asia | Southern Asia | Sushil Koirala |
| Turkey | Asia | Western Asia | Recep Tayyip Erdoğan |
| Iraq | Asia | Western Asia | Fuad Masum |
| Saudi Arabia | Asia | Western Asia | Salman of Saudi Arabia |
| Yemen | Asia | Western Asia | Abdrabbuh Mansur Hadi |
| Russia | Europe | Eastern Europe | Vladimir Putin |
| Ukraine | Europe | Eastern Europe | Petro Poroshenko |
| Poland | Europe | Eastern Europe | Andrzej Duda |
| Romania | Europe | Eastern Europe | Klaus Iohannis |
| Czech Republic | Europe | Eastern Europe | Miloš Zeman |
| Hungary | Europe | Eastern Europe | János Áder |
| Belarus | Europe | Eastern Europe | Alexander Lukashenko |
| Bulgaria | Europe | Eastern Europe | Rosen Plevneliev |
| United Kingdom | Europe | Northern Europe | David Cameron |
| Sweden | Europe | Northern Europe | Stefan Löfven |
| Italy | Europe | Southern Europe | Sergio Mattarella |
| Spain | Europe | Southern Europe | Mariano Rajoy |
| Greece | Europe | Southern Europe | Prokopis Pavlopoulos |
| Portugal | Europe | Southern Europe | Marcelo Rebelo de Sousa |
| Germany | Europe | Western Europe | Joachim Gauck |
| France | Europe | Western Europe | François Hollande |
| Netherlands | Europe | Western Europe | Mark Rutte |
| Belgium | Europe | Western Europe | Charles Michel |
| Austria | Europe | Western Europe | Heinz Fischer |
| Switzerland | Europe | Western Europe | Simonetta Sommaruga |

Table 8: List of Countries and Presidents Used in Our Experiment

