# OpenReview forum: "Multi-FAct: Assessing Factuality of Multilingual LLMs using FActScore"
_colmweb.org/COLM/2024/Conference — COLM_

### Official Review · Reviewer_bkdk · 2024-05-09

**Rating:** 7
**Confidence:** 4
**Ethics Flag:** 1

**Summary:**

This paper proposes to extend FACT-Score to be multilingual and shows various analyses across languages and geographies. They find that shorter Wikipedia pages in other languages provide less information to use for FACT-Score resulting in lower scores. They show this can be partially resolved through translation and ensembling of Wikipedia pages. They also show a western bias in FACT-Score.

My score is marginally below as of right now, but I think with fixing the grammar and other issues it would go to marginally above.

EDIT: increased my score, see response to their rebuttal.

**Questions To Authors:**

Minor comments:
1. The related work should be a cite paper with parentheses for many of the citations there (some are correctly using the name, others should be in parens).  This should also be followed throughout the paper - only use the non-parens version when citing the name directly.
2. 5.2 should be low resource language**s**
3. Footnote 5 on page 6 has an undefined reference
4. Table 3 could be made clearer by multiplying the values by 100 and underlining less - it is a little busy

**Reasons To Accept:**

- The paper provides many interesting analyses into how different Wikipedia languages play a role in impacting FACT-Score values.
- They propose several ways of countering this length bias that are intuitive and seem to help. Figures are well done and easy to understand
- The topic is important when discussing the evaluation of LLMs across languages

**Reasons To Reject:**

1.  Some of the experimental settings are unclear. For Table 3 and Figures 6-7, which references are being used? How many instances of data are used per language?  Will these datasets be open-sourced?
- Other questions include, how does each of the changes you made in terms of using open models impact these results? E.g. Does using Mistral cause a change?

2. There are a decent amount of mistakes in the paper in terms of spelling and latex references missing, which are minor and can be resolved but indicate a hurried submission

3. The paper title seems to indicate they extended Fact-Score itself, but in reality they applied it to new data - which is a fine contribution but somewhat misleading since there are no methodological contributions to the base FACT-Score.

---

> ### Author Rebuttal · Authors · 2024-05-31
>
> Thank you for comments and reviews.
>
> On the points mentioned under “Reasons to Reject”, here are our thoughts:
> - Regarding our experimental setting,
>   - We used 80 instances per language, with references from the English Wikipedia. All datasets and code will be open-sourced and made publicly available on GitHub.
>   - Table 3 shows the results of our experiment comparing the original InstructGPT implementation with our modified Mistral implementation. The use of Mistral resulted in a 2% point increase in performance compared to InstructGPT (originally reported in FActScore paper). We will ensure that the experimental details are clearly explained in the manuscript.
> - We appreciate the feedback about the title and understand that it may have been misleading. We had no intention to overstate our contributions. We will revise the title to “MultiFAct: Multilingual Factuality Evaluation using FActScore” and related sentence in the abstract to “we apply FActScore to multilingual settings.”
> - Thank you for the detailed feedback on the areas that need improvement. We apologize for the hasty writing and will tidy up overall manuscript. We will accept all the suggestions provided in the minor comments. To demonstrate how we would revise the paper to be more clear and concise, we have revised the first two paragraphs of Section 4.2, "Effect of Reference Length," as follows:
>
>    We hypothesize that non-English Wikipedias result in higher errors compared to English Wikipedia due to the lower coverage of topics, as evidenced by the shorter length of non-English articles. Figure 3b shows that there is a linear relationship between FActScore estimation error and the word count of Wikipedia articles, indicating that longer Wikipedia articles tend to serve better as references for factuality evaluation.
>   To further test the linear relationship observed, we select 100 individuals from each of our studied languages who have both English Wikipedia articles and substantially long Wikipedia articles in their regional language. We generate biographies for these individuals in English and evaluate them against both the English and regional language versions of Wikipedia. Since FActScore estimation for English is nearly perfect, we use it as a benchmark to estimate the error when using non-English articles as references. Figure 4a illustrates that the linear relationship between the length of articles and FActScore holds for longer articles as well.

---

> > ### Comment · Reviewer_bkdk · 2024-06-03
> > **Thank you for the response - increasing my score**
> >
> > I like this work and think it is an incorrect topic. The authors have promised some revisions which I think are minor and easy to do.
> >
> > I think the paper will make for good conversation at the conference and agree it is an understudied area.
> >
> > Re: other reviewer concerns: I agree that this paper doesn't propose any new methods and the evaluation process is simple. However, I don't think that is a major blocker - simple analyses done well are important also.
> >
> > I agree with reviewer a4f9 that more models would be nice - perhaps the authors could use Together.ai (if they can't run them locally) to use some of the larger models like Llama3-70B (but of course you couldn't have put that in the paper since it wasn't out yet).  I think the authors could and probably should do this so that there are reproducible results but again this wouldn't take very long to add.

---

### Official Review · Reviewer_5m3n · 2024-05-10

**Rating:** 4
**Confidence:** 4
**Ethics Flag:** 1

**Summary:**

This paper the problem of factuality evaluation of multilingual LLMs. It introduces a pipeline, Multi-Fact, for multilingual factuality evaluation, adapting FactScore for diverse languages. This pipeline first translates texts in other language into English, and then using FactScore for evaluation. In experiments, this paper explores the potential and limitations of using non-English resources for factuality evaluation of long-form generation. The Multi-Fact tool can also be used to investigate cultural and geographic biases in LLMs.

**Questions To Authors:**

N/A

**Reasons To Accept:**

- Factuality evaluation in multi-lingual LLMs is an interesting and important research problem.
- The proposed method is simple yet effective.
- There are some interesting findings in the experiments, which are inspiring for future research on building multi-lingual LLMs.
- The paper is well-written, and the idea is easy to follow.

**Reasons To Reject:**

- The novelty and technical complexity of the proposed method are quite basic. It mainly translates texts from other languages into English and then uses the FactScore tool for evaluation. There is no innovative approach or model created specifically for multilingual scenarios.
- The experiments do not compare the proposed method with any baseline methods. It is unclear whether the proposed method is effective, even though it seems reasonable.
- The experiments only use Wikipedia as a data source, which limits how widely the proposed method can be applied.

---

> ### Author Rebuttal · Authors · 2024-05-30
>
> Dear Reviewer 5m3n,
>
>
> Thank you for your review and comments.
>
>
> As to the concerns you have raised in the Rejections to Reject section, it may be helpful for us to clarify the main novelty and contribution of our paper. Please read below and let us know if you have additional questions or comments:
>
> * Yes, it is true that the technical complexity of the proposed method is relatively simple, translating texts from other languages to English and then applying  FactScore to evaluate them.
>
>     * The core contribution of this work is proposing a methodology that develops upon FactScore, which was only applicable in English, that can measure the factuality of text generated by multiple languages.
>     * We show that this methodology works for 8 different languages, which opens up possibilities to study the factuality of LLMs in many non-English languages. Our method is simple and easy to use, which makes it more powerful.
>
>     * We also rigorously tested that the translation step does not hurt the estimation (Section 3.4), and show how current LLMs are not performing differently for different languages, which is problematic.
>
> * About your concern regarding baseline: would you please explain a little more about what baseline we should have included?
>
>     * While we agree with you that comparison with baselines is important to show the validity of a metric, we did not include experiments comparing with baseline since our work is the first work that tries to evaluate multilingual models’ factuality in long form generation. Our factuality evaluation pipeline is based on FActScore (for English only) and we used the best-performing long-form factuality evaluation method.
>
>     * We will conduct an experiment by simply doing everything in the original language (without translation), as we think this is a valid baseline. If you are thinking of another, more valid baseline, it would be very helpful if you can tell us your idea.
>
> * About the use of Wikipedia data:
>
>
>     The proposed method is general and can be applied using any ground truth corpus with reasonably good coverage. None of the steps in our pipeline is dependent on the corpus being Wikipedia. In fact, you can use any corpus, in any language, into our pipeline.
>
>
>     We use Wikipedia because a previous study (Min et al 23) has shown it to be of reasonably good coverage in agreement with works (Chen et al., 2017;Petroni et al., 2021) that treat Wikipedia as a general-purpose knowledge corpus.

---

> ### Comment · Reviewer_5m3n · 2024-06-04
> **Reply**
>
> Thanks for your response. A few of my concerns have been addressed, but my major concern is the novelty of the method. Therefore, I will maintain my score.

---

> > ### Author Response · Authors · 2024-06-07
> > **Baselines**
> >
> > Dear Reviewer 5m3n,
> >
> > Here is the comparison with different baselines. Each column shows the difference with the ground truth (human evaluation) of each method.
> >
> > Trivial baselines:
> > Random prediction: 8.86%
> > Always "Supported": - 41.15%
> > Always "Not Supported": 58.86%
> >
> > | Method                     | Arabic | Swahili | Bengali | Chinese | Korean | French | German | Spanish | Mean   |
> > |----------------------------|--------|---------|---------|---------|--------|--------|--------|---------|--------|
> > | No Retrieval ChatGPT             | -26.83%| -28.15% | -25.97% | -26.48% | -27.70%| -27.75%| -28.82%| -28.95% | -27.58% |
> > | Retrieval + ChatGPT | -16.32% | -19.00% | -16.46% | -16.81% | -17.67% | -17.73% | -17.53% | -18.72% | -17.53% |
> > | Retrieval + Mistral        | -15.04%| -18.03% | -16.88% | -16.16% | -17.19%| -17.07%| -17.51%| -17.37% | -16.91% |
> > | NPM                        | -9.74% | -13.27% | -10.52% | -9.60%  | -9.81% | -14.00%| -14.87%| -15.25% | -12.13% |
> > | Retrieval + Mistral + NPM  | 2.80%  | -1.57%  | 2.19%   | 2.89%   | 0.63%  | -1.26% | -1.27% | -1.15%  | 0.41%  |
> >
> > * ChatGPT refers to GPT3.5-turbo.
> > * No Retrieval ChatGPT means simply asking ChatGPT whether a fact is correct or not (closed book setting)
> > * Retrieval + ChatGPT implies retrieving relevant passages from Wikipedia and then answering whether a fact is correct or not.
> > *Retrieval + Mistral refers to retrieving passages and then asking Mistral 7B for judgement
> > *NPM refers to Non Parametric Model, which requires having access to the Wikipedia article
> > *The final method is the ensemble version of NPM and Retrieval + Mistral and it performs the best. This is the one we used throughout our paper for other analyses.
> >
> > From the table, it is clear that, in order to reach good performance, an ensembling of NPM (Non Parametric Language Modeling (Min et al ‘22) and retrieval based method is critical. Retrieval + LLM usually overestimates model’s factscores and having NPM counteracts the effect of overestimation on average (similar conclusion was also reported in the original paper).
> >
> > Similarly, in our comment to reviewer r8Mi, we discuss in length about the difficulty and impracticality of fair factuality evaluation in a monolingual setting. The translation step seems critical if we want to disentangle a fair comparison of models’ generation in different languages.
> >
> > Our method, though very simple, and does not contain any novel model development, is still very effective for factuality evaluation in multilingual settings, which was our original goal.

---

### Official Review · Reviewer_r8Mi · 2024-05-21

**Rating:** 7
**Confidence:** 4
**Ethics Flag:** 1

**Summary:**

It is nice that this paper tackles the multilingual challenge – this is very, very underexplored in our field. Overall, the paper makes a few design decisions that I would have probably done differently, but the authors do justify their decisions relatively well and I think I would be able to replicate their methods. In particular, I think the paper could be improved with two experiments:
•	Including at least one small experiment using a non-LLM translation compared to a translation LLM.
•	Trying to see how FActScore would work on a language without translation. Aka an experiment similar to section 4.1, but just done monolingually in the languages (such as Bengali generation and Bengali evaluation).

One of my first questions when I read the paper was “Why translate with GPT-3.5?”. The authors do address this. It is ok for some languages, but if you are going to do automatic evaluation, another commercial solution like Google Translate may make more sense – especially for low-resource languages (Kocmi et al, 2023; Robinson et al., 2023). The authors make a nice point about gender in Google Translate and why they did not use it, but I would have still liked to have at lease seen one experiment showing what happens if you use that as well. A comparison even for one language with Google Translate and GPT-3.5.


I would have also liked to have seen how the method works monolingually (aka not translating to English). Again, the authors make claims as to why they made these decisions such as “which makes it difficult to compare cross-lingual factuality if the knowledge base is different.” However, it would be nice to still see how it works in different resource settings. You are generating in that language, it could be another line or two in some tables. This feels like it could also answer some of my questions about translationese impacts.

Overall, this paper is a nice attempt at multilingual fact checking, which is woefully underexplored. This alone is a worthwhile contribution to the field.

**Questions To Authors:**

Figure 5, the x-axis should be the same between a and b. Either (0.0-1.0) for both or (0.0-0.8) for both - not for 0.8 for one and 1.0 for the other.

**Reasons To Accept:**

There are very few papers looking at multilingual fact checking in LLMs. The field needs more of these.

**Reasons To Reject:**

I'd like to see the following two experiments:
•	Including at least one small experiment using a non-LLM translation compared to a translation LLM.
•	Trying to see how FActScore would work on a language without translation. Aka an experiment similar to section 4.1, but just done monolingually in the languages (such as Bengali generation and Bengali evaluation).

---

> ### Author Rebuttal · Authors · 2024-05-30
>
> Dear Reviewer r8Mi,
>
>
> Thank you very much for your long, detailed and insightful review! We agree on your opinion on translationese and impacts of doing factuality verification in English.
>
>
> On the matter of concerns listed under Reasons to Reject:
>
>
> * About non-LLM based translations: Thanks for pointing this out. It is our oversight not to include an experiment on the effects of different types of translation methodology in the appendix. We are conducting the experiment right now and will share the results with you in this thread as soon as we are done.
>
> * About the experiment doing FActScore monolingually:
>     * This is a very valid point, and the reason we didn’t do it before was that our method uses Mistral for atomic fact generation and then doing RAG on wikipedia to verify the validity of the facts. None of these two steps are possible by Mistral, since it is also English only. Besides, there is was concern regarding which languages’ corpus to use for a fair comparison across languages.
>
>
>     * However, provided that you asked, we are conducting the experiment on a subset of the languages using InstructGPT/GPT3.5-Turbo [which are multilingual]. We will let you know the experiment results in this thread as soon as it is done.
>
> If you have any more questions or concerns, please let us know!

---

> > ### Comment · Reviewer_r8Mi · 2024-06-06
> >
> > Thanks. I'm interested to see what these experiments show.

---

> > > ### Author Response · Authors · 2024-06-07
> > >
> > > Dear Reviewer r8Mi,
> > >
> > > The results for the two experiments you asked for are ready:
> > >
> > > 1. **About non-LLM based translation**:
> > > We did an ablation on the effect of LLM based translation and the results are as follows. We used a subset of the data provided in the original paper. The experiment set up is same as Table 1 in the paper and we just swapped GPT3.5 translation with Google translation:
> > >
> > > | Language | Error with GPT3.5 | Error with GTranslate |
> > > |----------|-------------------|-----------------------|
> > > | Arabic   | 2.80%             | 5.10%                 |
> > > | Swahili  | -1.57%            | 1.68%                 |
> > > | Bengali  | 2.19%             | 5.70%                 |
> > > | Chinese  | 2.89%             | 8.83%                 |
> > > | Korean   | 0.63%             | 4.53%                 |
> > > | French   | -1.26%            | -1.06%                |
> > > | German   | -1.27%            | -0.19%                |
> > > | Spanish  | -1.15%            | 5.81%                 |
> > > | **Mean** | **0.41%**         | **3.80%**             |
> > >
> > > In conclusion, GPT3.5 based translation is doing better than Google Translation for most of the languages, however, it seems like Google Translation is also *reasonably* good. Which means, in case we do not have access to LLM based translation, commercial translation like Google Translate will also work.

---

> > ### Author Response · Authors · 2024-06-07
> > **About Atomic breakdown and factuality evaluation in non-English language:**
> >
> > **About Atomic breakdown and factuality evaluation in non-English language:**
> >
> > Accurate evaluation of the fact verification system (using non-English atomic breakdown) in non-English language is difficult for several reasons:
> >
> > * The original NPM implementation (which is crucial for reaching good accuracy)  was trained on English data only and the tokenizer heavily disfavors non-English languages, which is why running the most effective method (ensembling with NPM) in non-English language is not possible [refer to the table at the very end of this comment]. Implementing non English NPM is a worthwhile objective, but out of scope from our project.
> > We, thus, decide to only include results for Retrieval + ChatGPT method for doing atomic fact breakdown in non-English language. [We do not provide numbers for Mistral since Mistral is English only model]
> >
> > * Besides, to make a fair comparison between two languages’ factuality, while doing atomic breakdown in the corresponding language, we need to have the same ground truth corpus in both languages.
> > To meet the later requirement, we translate wikipedia in the non-English language to English (as X→ EN usually leaves out less translationese than EN→X)
> >     * This setting is similar to Figure 4a. We sampled 10000 wikipedia articles from French and Korean wikipedia and then took the top 120 articles by length and asked GPT4 to generate biographies corresponding to those entities. We translate those 120 wikipedia articles from KOR/French into English for a comparison.
> >
> > * Since we do not have ground truth labeled data for these 120 entities in French/Korean, we only show the difference with the best performing method (Retrieval + Mistral + NPM using EN wikipedia) here (like Figure 4a)
> >
> >
> > We only conducted this experiment for Korean and French (one latin and one non-latin script language)
> >
> > | Language | Wiki-X-breakdown-in-X | X-wiki-translated-to-EN-breakdown-in-EN |  original-EN-wiki_breakdown-in-EN |
> > |----------|----------------------------------|--------------------------------|-----------------------------------------|
> > | French   |-20.28                         | -25.46                                  | -26.49                           |
> > | Korean   | -17.36                         | -34.27                                  |-35.86                           |
> >
> > The column represents the difference with the best performing method (Retrieval + (Mistral + NPM)) with the column. The method is same for all the cases (Retrieval + ChatGPT):
> > * **Wiki-X-breakdown-in-X:** We used atomic breakdown in KOR/FR and also verified against the corresponding wikipedia
> > * **X-wiki-translated-to-EN-breakdown-in-EN:** We did atomic breakdown in English, and verified against the translated KOR/FR wikipedia, that has been translated into English
> > * **original-EN-wiki_breakdown-in-EN:** Atomic breakdown in English and then verified against English wikipedia
> >
> > **In conclusion:**
> > * It is difficult to accurately quantify whether doing atomic breakdown in non-English languages will yield good results or not given the large difference with the best performing method.
> > * Still, it is possible that if there is a multilingual NPM model, the accuracy would go higher.
> > * A qualitative look at the atomic breakdown of biographies generated in Korean shows that the generated facts are accurate. This suggests that once we have a good multilingual NP model, our method would work well in the Korean setting.
> >
> >
> > The methodology comparison table (atomic fact breakdown in English):
> >
> > | Method                     | Arabic | Swahili | Bengali | Chinese | Korean | French | German | Spanish | Mean   |
> > |----------------------------|--------|---------|---------|---------|--------|--------|--------|---------|--------|
> > | No Retrieval ChatGPT             | -26.83%| -28.15% | -25.97% | -26.48% | -27.70%| -27.75%| -28.82%| -28.95% | -27.58% |
> > | Retrieval + ChatGPT | -16.32% | -19.00% | -16.46% | -16.81% | -17.67% | -17.73% | -17.53% | -18.72% | -17.53% |
> > | Retrieval + Mistral        | -15.04%| -18.03% | -16.88% | -16.16% | -17.19%| -17.07%| -17.51%| -17.37% | -16.91% |
> > | NPM                        | -9.74% | -13.27% | -10.52% | -9.60%  | -9.81% | -14.00%| -14.87%| -15.25% | -12.13% |
> > | Retrieval + Mistral + NPM  | 2.80%  | -1.57%  | 2.19%   | 2.89%   | 0.63%  | -1.26% | -1.27% | -1.15%  | 0.41%  |
> >
> >
> > **Notes:**
> > * ChatGPT refers to GPT3.5-turbo.
> > * No Retrieval ChatGPT means simply asking ChatGPT whether a fact is correct or not
> > * Retrieval + ChatGPT implies retrieving relevant passages from Wikipedia and then answering.
> > * Retrieval + Mistral refers to retrieving passages and then asking Mistral 7B for judgement
> > * NPM refers to Non Parametric Masked Language Modeling (Min et al ‘22), which requires having access to the Wikipedia article
> > * The final method is the ensemble version of NPM and Retrieval + Mistral and it performs the best. This is the one we used throughout our paper for other analysis.

---

### Official Review · Reviewer_a4f9 · 2024-05-22

**Rating:** 5
**Confidence:** 3
**Ethics Flag:** 1

**Summary:**

This paper explores multilingual LLM evaluation by adapting existing factual evaluation method FActScore. The paper also studies the potential and limitations of using non-English resources such as non-English wikipedia and geographic biases in LLM.

**Reasons To Accept:**

The paper adapts FActScore to a multi-lingual factuality evaluation framework, and experiment the method using 8 languages

Providing insight about reasonably large non-English wikipedia articles can also serve as a viable source.

**Reasons To Reject:**

Overall, the originality of the work is limited, it uses GPT3.5 for language translation then adapts an existing factuality method.

The paper can be improved by providing more details about FActScore, it is hard for readers to understand the approach with no knowledge of the method.

Experiments only use GPT3.5 and GPT4 for evaluation, it is better to have more baseline methods such as llama, mistral, etc.

---

> ### Author Rebuttal · Authors · 2024-05-30
>
> Dear Reviewer a4f9,
>
>
> First of all, thanks for your comments and reviews.
>
>
> On the points mentioned under “Reasons to Reject”, here are our thoughts:
>
> * It is true that our method uses an existing factuality evaluation method after translating multilingual generations into English. We made this choice for several reasons:
>     * Most LLMs, and all open-source LLMs, are not able to follow instructions properly in a multilingual setting, which means much of the original FActScore pipeline (which strongly depends on models’ capability to break down sentences into atomic facts and then do RAG) is unusable in other languages. This means it is simply impossible to evaluate and compare long-tail languages in a scalable fashion without translating them to English with current models.
>     * One of the main contributions of our paper (Section 3.4) was to show that we can use existing English factuality evaluation methods, like FActScore for non-English languages by translating the generations to English, which is not trivial at all. In fact, many tasks and metrics for LLMs that are originally built for English are not transferable to other languages without significant additional research, and it is unfair to ask for technical novelty on top of the 9 different languages (including low-resource languages) that we included in this work. Section 3.4.2 shows that the translation step does not significantly hurt factuality evaluation.
>
> * About providing more details on the FActScore details: We will modify our writing and the main figure accordingly so that it is easy for new readers to understand our method. Thanks for pointing this out!
>
> * On the matter of only using GPT4 and GPT3.5 for evaluation: Our method uses GPT3.5 based translation and Mistral 7B for factuality measurement. We did not evaluate llama, mistral (or any other open-source models) since they are meant to be English-only models and are not able to follow instructions/generate properly well for the languages we worked on. Most of our experiments are conducted on GPT3.5 and GPT4 because those two were the only strong multilingual models available at the time of writing the paper. More multilingual models like Claude 3 and Gemini 1.0 came out much later and they are also included in the appendix (Table 5). However, we plan to include models that came out since we submitted our paper (Command R+ and Gemini 1.5) in our final version of the paper.

---

### Decision · Program_Chairs · 2024-07-10

**Decision:**

Accept

**Comment:**

This paper uses FactScore to perform an evaluation of factuality in multiple languages. To do, so it generates responses in non-English languages, translates them to English, then runs FactScore. It also replaces several of the components of FactScore with open source models.

The reviews on this paper were split, two positive and two negatives. Reviewers appreciated the topic of multilingual fact checking, and the interesting analyses of the effect of different Wikipedias. There were questions about the effect of translating into English, and although the authors already investigated this in their original paper, they added more analyses, including translating with another translation software and performing some factuality evaluation in non-English languages. These additions greatly improve confidence in the validity of the results and should be included in any revision.

The main concerns raised by the reviewers were the limited novelty and missing baselines. While the paper doesn't propose a new method, I agree with the authors and one of the reviewers that this isn't a problem -- the investigation and results are interesting and useful to the community as they are. Concerning missing baselines, the author made an attempt to add a simple random baseline and provided ablations for their method (https://openreview.net/forum?id=lkrH6ovzsj&noteId=S3yAsA5YMt), which are helpful. These, too, should be included in any revision.

Given the improvements to the paper, and the valuable topic and insights, I believe this paper provide a useful contribution to the community and should be accepted, on the assumption that the above comments are addressed for the next revision.

[At least one review was discounted during the decision process due to quality]